# Simple Disposable Odor Identification Tests for Predicting SARS-CoV-2 Positivity

**DOI:** 10.3390/ijerph181910185

**Published:** 2021-09-28

**Authors:** Laura Ziuzia-Januszewska, Paweł Dobrzyński, Krzysztof Ślączka, Jaromir Ciszek, Łukasz Krawiec, Waldemar Wierzba, Artur Zaczyński

**Affiliations:** 1Department of Otolaryngology, Central Clinical Hospital of the Ministry of the Interior and Administration, 02-507 Warsaw, Poland; pawel.dobrzynski@cskmswia.gov.pl (P.D.); krzysztof.slaczka@cskmswia.gov.pl (K.Ś.); jaromir.ciszek@gmail.com (J.C.); lukasz.krawiec@cskmswia.gov.pl (Ł.K.); 2UHE Satellite Campus, University of Humanities and Economics, 01-513 Warsaw, Poland; waldemar.wierzba@cskmswia.gov.pl; 3Central Clinical Hospital of the Ministry of the Interior and Administration, 02-507 Warsaw, Poland; artur.zaczynski@cskmswia.gov.pl

**Keywords:** olfactory, anosmia, COVID-19, objective, SDOIT

## Abstract

Olfactory dysfunction (OD) is a common manifestation of COVID-19 and may be useful for screening. Survey-based olfactory evaluation tends to underestimate the prevalence of OD, while psychophysical olfactory testing during a pandemic has the disadvantage of being time consuming, expensive, and requiring standardized laboratory settings. We aimed to develop a quick, simple, affordable, and reliable test to objectively assess the prevalence and diagnostic accuracy of OD in COVID-19. The olfactory function of 64 COVID-19 inpatients and 34 controls was evaluated using a questionnaire and a simple disposable odor identification test (SDOIT) developed for this study. Four SDOIT models were assessed: 10-SDOIT, 9-SDOIT, 8-SDOIT, and 4-SDOIT, with 10, 9, 8 and 4 samples, respectively. We found a high frequency of self-reported OD in COVID-19 patients, with 32.8% and 42.2% reporting current and recent OD, respectively. Different SDOIT models revealed smell impairment in 54.7–64.1% of COVID-19 patients. The combination of either 10-SDOIT results and self-reported OD, or 8-SDOIT results and self-reported OD, were the best predictors of COVID-19, both with an AUC value of 0.87 (0.85 and 0.86 for the age-matched subjects). OD is a common symptom of COVID-19. A combination of self-reported smell deterioration and OD psychophysically evaluated using SDOIT appears to be a good predictor of COVID-19.

## 1. Introduction

A growing body of evidence shows a high incidence of olfactory dysfunction (OD) in coronavirus disease 2019 (COVID-19), with prevalence ranging from 5 to 98.3% [1,2,3,4,5,6,7]. Therefore, it has been hypothesized that new-onset smell impairment could serve as a potential predictor of SARS-CoV-2 infection [8,9].

Most of the previous studies on OD in COVID-19 are survey-based. However, self-assessment of OD tends to underestimate its true prevalence due to recall bias and subjects not being aware of their smell impairment [2,10], especially while experiencing other, severe symptoms such as respiratory distress [11]. Indeed, Moein et al. [2] found that 98% of COVID-19 patients exhibited OD when assessed objectively, compared with only 28% self-reporting smell deterioration. Similarly, in a study by Vaira et al. [12], objective evaluation revealed mild hyposmia in 30.3% of subjectively normosmic patients. Moreover, several meta-analyses showed a higher overall prevalence of OD when using objective compared with subjective assessment methods (72.1–77% vs. 44.5–53%, respectively) [11,13,14].

However, psychophysical olfactory tests are time consuming, expensive, and require standardized laboratory settings, and thus, they are difficult to perform during a pandemic [13]. Moreover, disposable tests are preferable to reduce the risk of viral contamination [15]. Self-administered home-based objective olfactory tests have been proposed [16,17,18]. However, odor identification evaluation in these settings may be less reliable as subjects ought not know which odorants are being tested [16], and the olfactory threshold assessment or intensity ratings that are proposed in the literature [16,17,18] may be more difficult to apply than performing a relatively simple identification test. Nevertheless, these novel methods appear to be valuable alternatives to the standard psychophysical tests.

In the present study, we aimed to psychophysically evaluate the prevalence of OD in 64 hospitalized patients with laboratory-confirmed COVID-19 and to develop a quick, simple, affordable, and reliable test to screen for SARS-CoV-2 infection. Consequently, we propose a simple disposable odor identification test (SDOIT).

## 2. Materials and Methods

### 2.1. Subjects and Settings

This case-control study was conducted between April 2020 and August 2020 at the Central Clinical Hospital of the Ministry of the Interior and Administration in Warsaw, Poland, which has been designated by the Government for the treatment of patients suffering from COVID-19. The inclusion criteria for the case group were adults (≥18 years old) with laboratory-confirmed SARS-CoV-2 infection. COVID-19 was diagnosed by RT-PCR performed on nasopharyngeal samples, utilizing SARS-CoV-2 nucleic acid detection kit (GeneFinder TM COVID-19 Plus RealAmp Kit) in adherence to the protocols supplied by the kit manufacturer. The control group consisted of healthy adult volunteers with no symptoms of COVID-19 (other than recent OD, which was not queried for to avoid selection bias) or upper respiratory tract infection. Although a negative test result for SARS-CoV-2 was not an inclusion criterion for the control group, the total number of individuals diagnosed with COVID-19 from the beginning of the pandemic to the end of August 2020 was lower than 0.18% of the Polish population (67372/38354000) [19], hence the risk of infection in asymptomatic volunteers was considered to be very low.

The exclusion criteria were age below 18 years of age, pregnancy, a history of pre-existing OD, head trauma, rhinosinusitis, or other chronic nasal disease, and inability to complete the questionnaire due to a history of neurocognitive disorders, an altered state of consciousness, or the need for intensive care and / or invasive ventilation at the time of the survey.

### 2.2. Clinical Outcomes

All participants completed a questionnaire regarding: (1)General demographic data;(2)Medical history (comorbidities, chronic medication use, tobacco addiction, and pre-existing OD);(3)COVID-19 course (date of first symptoms, nasal, and general symptoms), and;(4)Olfactory function—participants rated their sense of smell at its worst since the onset of the disease (“recent OD”) as “normal”, “decreased”, or “none at all”, as well as using the visual analogue scale (VAS), from 0 (normal sense of smell) to 10 (no sense of smell).

The onset (in relation to the day of the survey and to other COVID-19 symptoms, expressed as either before, or concurrently, or after) and persistence (complete, or incomplete, or no recovery, or worsening) of OD were evaluated. As none of the subjects reported an incomplete recovery, the olfactory function at the time of the survey (“current OD”) for the patients reporting complete recovery was classified as “normal” and “0” on the descriptive scale and VAS, respectively, and for the remaining patients classified as equal to the “recent OD”. Any data missing from the forms and information regarding the course of the disease were transcribed from the electronic medical records. The physiological parameters were assessed at least once a day using the modified early warning score (MEWS) [20] adapted by the hospital therapeutic committee by including oxygen saturation and need for oxygen supplementation. 

### 2.3. Psychophysical Evaluation

The psychophysical olfactory evaluation was performed concurrently with the questionnaire on all participants, using the simple disposable odor identification test (SDOIT) that we had developed for the purpose of this study. Ten disposable test paper strips numbered 1 to 10, 9 of which contained well-known pure odorants (commercially available cinnamon, mint, lemon, coffee, clove, rose, anise and camphor essential oils and disinfectant alcohol) and the remaining 1 with an odorless control (deionized water) were each enclosed in plastic covers so the odors did not mix. Upon removal of the covers, each odor was presented to a patient at 30-second intervals to prevent olfactory desensitization. The odorants used in the study were selected to include both unimodal odors, with little or no trigeminal stimulation, and bimodal odors, with mixed stimulation of the olfactory and trigeminal nerve. For each odor strip, patients were asked to indicate whether they detected an odor and if so, to identify the odor (using a forced choice format, with 4 given options per test odorant; see Table 1). The SDOIT test kit is shown in the Appendix A. We defined OD in the SDOIT when a patient correctly identified a number of odors lower than the 10th percentile of the results from the control group, as performed by Iravani et al. [16], and as commonly used in other previous olfactory tests [10,16]. This resulted in defining OD as at least 2 incorrect answers in 10-SDOIT, 9-SDOIT and 8-SDOIT and at least 1 incorrect answer in 4-SDOIT. We did not assess other components of olfaction such as odor threshold and odor discrimination, so that we could maximize the simplicity and minimize the time necessary for the assessment. 

### 2.4. Ethical Concerns

This study was approved by the Ethics Committee of the Central Clinical Hospital of the Ministry of the Interior and Administration in Warsaw and was performed in accordance with the ethical standards of the Declaration of Helsinki and its later amendments. Voluntary written informed consent was obtained from all participants. All tests were performed with the highest regard for patients’ and examiners’ safety using appropriate personal protective equipment.

### 2.5. Statistical Analyses and ROC Analysis of COVID-19 Predictors

Usual descriptive statistics were used, as shown in Table 2, Table 3, Table 4, Table 5, Table 6, Table 7 and Table 8. Fisher’s exact test was used to compare self-reported OD and the Mann–Whitney U test was used to compare the SDOIT results between cases and controls. To assess the correlation between the self-report olfactory function and SDOIT results, Fisher’s exact test and the Chi-square test were used. In cases, the Spearman correlation coefficient was used to test the correlation between clinical features and OD in the case where the studied variables were quantitative, the Mann-Whitney U test to study differences within a qualitative variable and a quantitative variable, Fisher’s exact test for two qualitative variables. Statistical analysis was performed with R software (version 3.6.0, R Foundation for Statistical Computing, Vienna, Austria). A level of *p* < 0.05 was used to determine statistical significance. The receiver operating characteristic (ROC) curves were plotted and the area under the ROC curve (AUC) analysis was performed to assess the utility of 19 selected classifiers (as described in the Results section below) in predicting SARS-CoV-2 positivity. The sensitivity, specificity, positive predictive value (PPV), and negative predictive value (NPV) of these predictors were also evaluated. For psychophysical test we assessed both the models defining OD as SDOIT scores below the cut-off value set at 10th percentile of the results obtained in the control group, and the models defining OD as scores below the optimal cut-off values calculated in ROC analysis. For the ROC analysis, for the combination of SDOIT results (with OD defined in each case as in the description of a given classifier) and the self-reported OD, a variable with three possible values was introduced: (1) subjective normosmia and normal SDOIT result (no OD in SDOIT); (2) subjective normosmia and OD in SDOIT; (3) subjective OD (it was not necessary to add more categories, as only cases reported subjective OD). We did not include other (non-smell-related) COVID-19 symptoms in the analysis because the role of additional symptoms could be overestimated due to selection bias, as the control group consisted of healthy individuals.

## 3. Results

### 3.1. Clinical Outcomes

A total 64 cases (29 women and 35 men; mean age, 52.3 ± 20.9 years) and 34 controls (26 women and 8 men; mean age, 40.9 ± 10.7 years) were included in the study. The most prevalent symptoms of COVID-19 (other than OD) were fatigue (70.3%; *n* = 45), cough (39.1%; *n* = 25), fever (37.5%; *n* = 24), headache (35.9%; *n* = 23), and gastrointestinal complaints (35.4%; *n* = 23). Nasal congestion was reported by 32.8% (*n* = 21), and rhinorrhea by 29.7% (*n* = 19) of those with COVID-19. The demographic and clinical characteristics are summarized in Table 2, Table 3, Table 4 and Table 5 and in the Appendix A.

### 3.2. Self-Reported Olfactory Function

The surveys and olfactory evaluations took place 6.9 ± 6.1 days (range 0–24) after the first positive RT-PCR result and 9 ± 7 days (range 1–30) from the reported onset of OD (note that 2 patients did not remember the time of onset of OD). There was no significant correlation between self-reported OD and the time between the positive PCR result and the questionnaire (*p* = 0.565). OD appeared before (14.8%, *n* = 4), simultaneously (11.1%, *n* = 3), or after (74.1%, *n* = 20) the presentation of other COVID-19 symptoms. At the time of the evaluation, 7 patients (25.9%) reported complete recovery of olfactory function, while 20 patients (74.1%) reported no recovery (55.6%; *n* = 15), or worsening of olfactory function (18.5%; *n* = 5).

The frequency of self-reported smell impairment was significantly higher in COVID-19 patients, with 32.8% and 42.2% of patients reporting current and recent OD, respectively, while all the control subjects reported normosmia. The mean VAS score for smell deterioration was significantly higher in the study cases compared with the controls. These results are presented in Table 6.

As there was a difference between cases and controls in terms of age, we have also analyzed self-reported OD and SDOIT scores for the subset of 75% youngest COVID-19 patients (*N* = 48) whose age did not differ significantly from the control group (*p* = 0.531). The inter-group differences remained highly significant, with 29.2% and 41.7% reporting current and recent OD, respectively (Table 7).

### 3.3. Psychophysical Evaluation

The psychophysical evaluation was performed for all the study subjects (*n* = 98). Mean percentages of correct answers (for all 10 samples, including the non-odorant sample), detected odors and identified odors in cases vs. controls were 65.6% vs. 95.8%, 83.5% vs. 100%, and 66.3% vs. 95.6%, respectively (Table 6). For all odors, lower p-values were achieved for identification than for detection. Considering the identification results for each odor separately, we created two additional, shortened psychophysical test models, one with 8 and the other with four selected odorants. Therefore, 4 SDOIT models were included for further analysis:10-SDOIT, evaluating the number of correct answers (correct identification of nine odors and reporting of no odor detection in an odorless sample);9-SDOIT evaluating the number of identified odors out of nine odorants;8-SDOIT evaluating the number of identified odors out of eight odorants (cinnamon, mint, lemon, coffee, clove, anise, camphor, and alcohol)—excluding odorant showing no significant differences between cases and controls (rose), and;4-SDOIT evaluating the number of identified odors out of four odorants (cinnamon, mint, lemon, and alcohol)—showing the highest intergroup differences (with *p* ≤ 0.001).

The results for the individual odor tests are presented in Appendix A.

In the intergroup comparison, the mean scores in all four models were significantly lower in cases than in controls (*p* < 0.001 for all models, Table 6). Taking the cut-off value at the 10th percentile of the results in controls, we found OD in 59.4% (38/64) vs. 8.8% (3/34), 54.7% (35/64) vs. 5.9% (2/34), 54.7% (35/64) vs. 2.9% (1/34) and 64.1% (41/64) vs. 5.9% (2/34) of cases vs. controls, for 10-SDOIT, 9-SDOIT, 8-SDOIT and 4-SDOIT, respectively. OD was significantly associated with SARS-CoV-2 positivity, with odds ratios (OR) of 15.1, 19.3, 39.8 and 28.5 for 10-SDOIT, 9-SDOIT, 8-SDOIT and 4-SDOIT, respectively.

In the comparison of controls and the subset of 75% youngest COVID-19 patients (*N* = 48) the differences remained highly significant, with mean percentages of correct answers (for all 10 samples), detected odors and identified odors in 75% youngest patients being 74.2%, 85.4% and 72.9%, respectively (Table 7).

There was a significant correlation between the current self-reported OD and psychophysically assessed OD in all the SDOIT models. (Table 8). Subjectively, normosmic COVID-19 patients showed OD at psychophysical evaluation in 51.2%, 46.5%, 46.5% and 55.8% of cases for 10-SDOIT, 9-SDOIT, 8-SDOIT and 4-SDOIT, respectively.

### 3.4. Correlations between OD and Patient Characteristics

There were no significant gender differences in self-reported olfactory function and psychophysical test results. Mean SDOIT scores were significantly lower among older patients, but there was no correlation between age and maximum VAS score and self-reported OD. Within the COVID-19 group, nasal obstruction was more prevalent in patients reporting OD compared with normosmic subjects (51.9% vs. 18.9%, and 52.4% vs. 23.3% for recent and current OD, respectively) and rhinorrhea was more prevalent in patients reporting recent OD (51.9% vs. 13.5%), but not current OD. No significant correlation was found between nasal symptoms and SDOIT scores. Patients with worse psychophysical test results had higher MEWS scores and were hospitalized longer, and 9-SDOIT and 8-SDOIT scores were significantly lower in patients who later died, while there were no correlations between the length of hospitalization, MEWS score and death with self-reported OD. The need for oxygen supplementation was less frequent in patients reporting recent OD, but more frequent in patients with lower SDOIT scores. The most important correlations between OD and patient characteristics are shown in Table 3, Table 4 and Table 5. More detailed data regarding these correlations are presented in the Appendix A.

### 3.5. Assessment of COVID-19 Predictors and ROC Analysis

We selected 19 classifiers for predicting SARS-CoV-2 positivity, including self-reported OD (with recent OD found to be a better classifier than current OD and therefore used in further assessments), maximum VAS score (with a cut-off point at <5), 4-SDOIT models and the combination of survey-based and psychophysical olfactory evaluation, as presented in Table 9. Self-reported recent OD achieved sensitivity of 42%, specificity of 100%, PPV of 100%, NPV of 48% and AUC of 0.71, and the maximum VAS score achieved sensitivity of 64%, specificity of 100%, PPV of 100%, NPV of 60% and AUC of 0.82 for predicting SARS-CoV-2 positivity. Our psychophysical evaluation, when defining OD as the score below the 10th percentile of healthy subjects, found 4-SDOIT to be the best classifier, with sensitivity of 64% and specificity of 94%, PPV of 95%, NPV of 63% and AUC of 0.8. However, the optimal cut-off point calculated in the ROC analysis for all SDOIT models was at least one incorrect answer with AUC of at least 0.8 for all models. The combination of SDOIT results and self-reported OD (with any OD, either subjective or objective, indicating COVID-19) improved the diagnostic accuracy. The inclusion of VAS did not improve these classifiers. To minimize the risk of patients suspected of infection eligible for isolation not being detected, classifiers with the highest AUC and the highest sensitivity were selected as the best predictors of COVID-19, combining self-reported OD and OD defined in SDOIT as:(1)0-9/10 correct answers in 10-SDOIT (with AUC of 0.87, sensitivity of 91%, specificity of 71%, PPV of 85% and NPV of 80%), and;(2)0-7/8 identified odors in 8-SDOIT (with AUC of 0.87, sensitivity of 86%, specificity of 79%, PPV of 89% and NPV of 75%).

The main results are presented in Table 9 and Figure 1. The detailed results of the ROC analysis are presented in Appendix A.

We have also performed the ROC analysis for controls and the subset of 75% youngest COVID-19 patients to eliminate the potential impact of age. These results are presented in Table 10 and Figure 2. In this analysis, self-reported recent OD achieved sensitivity of 42%, specificity of 100%, PPV of 100%, NPV of 55% and AUC of 0.71, and the maximum VAS score achieved sensitivity of 35%, specificity of 100%, PPV of 100%, NPV of 52% and AUC of 0.68 for predicting SARS-CoV-2 positivity. Similarly to the results of the analysis for the entire cohort, the analysis of age-matched group showed the 4-SDOIT to be the best classifier when defining OD as the score below the 10th percentile of healthy subjects, with sensitivity of 54% and specificity of 94%, PPV of 93%, NPV of 59% and AUC of 0.75, but the optimal cut-off point calculated in the ROC analysis for all SDOIT models was at least one incorrect answer with AUC of 0.76, 0.76, 0.78 and 0.75 for 10-SDOI, 9-SDOIT, 8-SDOI and 4-SDOIT, respectively. The combination of SDOIT results and self-reported OD improved the diagnostic accuracy. Selecting classifiers with the highest AUC and the highest sensitivity, the best predictors of COVID-19 were these combining self-reported OD and OD defined in SDOIT, i.e.,:(1)0-9/10 correct answers in 10-SDOIT (with AUC of 0.85, sensitivity of 88%, specificity of 71%, PPV of 81% and NPV of 80%), and;(2)0-7/8 identified odors in 8-SDOIT (with AUC of 0.86, sensitivity of 83%, specificity of 79%, PPV of 85% and NPV of 77%).

## 4. Discussion

We found self-reported recent OD in 42.2% of COVID-19 patients. This is consistent with pooled prevalence estimates for OD reported in meta-analyses, ranging from 35 to 56% [11,13,14,21,22]. Although many previous studies reported a higher prevalence of OD in women [4,5,6], several authors found no gender differences, especially when using objective tests [2,17]. Likewise, we did not observe any differences in the prevalence of OD based on sex. This may indicate that previously reported female predominance reflects an increased sensitivity of women in detecting chemosensory dysfunctions or their greater propensity to complete surveys [14,23].

Most of the studies regarding smell impairment in COVID-19 are survey-based, however, studies show that self-assessment of olfactory function tends to underestimate the prevalence of OD [2,10]. Using a SDOIT we found that, although there was a significant correlation between self-reported smell impairment and the psychophysical test results, OD was more frequently revealed by psychophysical evaluation (54.7–64.1% in different SDOIT models), than in the self-reported data (32.8% for current OD). It was noteworthy that in our study, among subjectively normosmic COVID-19 patients, approximately 50% showed OD at psychophysical evaluation. These results are consistent with the findings of the aforementioned studies [2,11,12,13,14] and highlight the importance of psychophysical smell assessment.

Many studies have shown anosmia to be associated with the mild course of COVID-19 [5,7,8,14,21,24]. However, other studies either failed to find this relationship [2,17] or reported smell impairment to be associated with severe forms of the disease [16]. We found that longer hospitalization, higher MEWS scores and, for some SDOIT models, death, indicative of severe illness, were associated with worse psychophysical test results, but not with the self-reported OD. Moreover, the need for oxygen supplementation was less frequent in patients reporting OD, but more frequent in subjects with lower SDOIT scores. This confirms a hypothesis that previously reported associations of OD and the mild course of the disease may be due to neglecting smell impairment by patients with severe respiratory distress and should not be considered as a positive prognostic factor [12,23].

Smell impairment usually occurs early in the course of COVID-19 [2,3,12] and may sometimes be the first or even the sole symptom of SARS-CoV-2 infection [7,12]. In our study, OD appeared before (14.8%), simultaneously (11.1%), or after (74.1%) the presentation of other COVID-19 symptoms, which is consistent with a study by Lechien et al. [5], who reported smell dysfunction occurring prior (11.8%), concomitantly (22.8%) and after (65.4%) the appearance of general or ENT manifestations. Moreover, many studies have reported the early recovery [4,6,12,17] of OD in cases of COVID-19. In our study, 25.9% of patients reported complete recovery of olfactory function, while 74.1% reported either no recovery (55.6%) or worsening of olfactory function (18.5%); however, the longest duration of smell impairment at the time of the survey was 30 days. Interestingly, Vaira et al. [17], observed that 80% of patients reporting complete recovery of chemosensitive functions revealed some residual abnormalities in objective testing. In contrast, in our study, almost all the subjectively recovered patients were normosmic upon psychophysical evaluation, with only one subject misidentifying one odor.

The early onset and early recovery of OD argues in favor of a conductive pathomechanism of COVID-19 related anosmia [7]. However, many COVID-19 patients have reported OD in the absence of nasal obstruction and rhinorrhea [3,17]. In our study, although nasal symptoms were significantly more prevalent in subjects reporting OD, they were absent in 40.7% (11/27) of these patients and were not associated with worse SDOIT scores. Hence, rhinitis and nasal congestion do not appear to be the main causative factors in COVID-19 related OD.

Gustatory disorders commonly observed in COVID-19 have been suggested to result from impaired flavour perception due to retronasal olfactory dysfunction [25,26]. However, some studies have shown dysgeusia to be more frequent than OD [17], and expression of ACE2 receptors at high levels has been found in the oral mucosa [27,28], suggesting a distinct pathomechanism [27]. Nevertheless, as true gustatory dysfunction is often difficult to distinguish from OD [3], we chose not to include it in our study.

Smell impairment has been found to be highly associated with SARS-CoV-2 positivity (OR > 10) [8,26], with high specificity (93–99%), but low-to-moderate sensitivity (23–48%) [11,29] and has even been assessed to be the strongest predictor of COVID-19 [8,9]. Similarly, in our study self-reported OD achieved specificity of 100%, sensitivity of 42% and AUC of 0.71 in predicting SARS-CoV-2 infection, for both the entire, and the age-matched subjects. According to Karni et al. [30], a quantitative smell assessment (1–10 scale) was even more effective, with 66% sensitivity, 97% specificity and 0.81 AUC. Similarly, we found that a maximum reported VAS with a cut-off point of five achieved a higher sensitivity (64%) and AUC (0.82), with specificity of 100%, indicating a better discriminatory ability in predicting COVID-19 compared with binary self-assessment of smell (normosmia vs. OD). However, in the analysis with the 75% youngest patients, the predictive value of VAS score was lower, with the AUC of 0.68. Huart et al. [31], found that an identification score of the extended “Sniffin’ Sticks” test battery showed good discrimination between COVID-19 patients and controls, with a 100% sensitivity and 80% specificity. In our study, the SDOIT models also had a good discriminating ability in predicting COVID-19 with AUC of 0.8–0.82, sensitivity of 64–80%, and specificity of 71–94% in the entire cohort and with AUC of 0.75–0.78, sensitivity of 54–85%, and specificity of 71–94% in the age-matched subjects. Moreover, the combination of SDOIT results and self-reported OD resulted in improved diagnostic accuracy with AUC of 0.85–0.87, sensitivity of 78–91%, specificity of 71–94%, PPV of 85–89%, and NPV of 70–80% in the entire cohort and with AUC of 0.84–0.86, sensitivity of 73–88%, and specificity of 71–94% in the age-matched subjects. These findings support the role of OD as the early marker of COVID-19 [9] and an indication for immediate isolation and laboratory testing, or even retesting when the first RT-PCR result is negative [32]. It was noteworthy that the sensitivity (91% and 88% for the entire cohort and the age-matched subjects, respectively) and NPV (80% in both cases) were highest for the 10-SDOIT-based model), while the specificity for the 8-SDOIT-based model (79%) was higher than for the 10-SDOIT-based model (71%). Hence, we suggest than when there is enough time and a satisfactory availability of RT-PCR assay, one should consider the combination of self-reported OD and 10-SDOIT; however, with limited time and resources, the combination of 8-SDOIT and self-reported OD seems to be adequate as a predictor of SARS-CoV-2 positivity and an indication for RT-PCR testing.

Our study had several limitations. First, our sample size was limited, and the results may be influenced by the single institutional nature of the study. To improve sampling, the data were acquired over quite a long period of time. Furthermore, the patients were assessed at different time periods following the onset of infection and some reported having already recovered. However, this may have led to the underestimation of OD prevalence and significance, rather than the opposite. Moreover, we did not assess the recovery pattern of OD. Future follow-up study should be considered. In addition, our test is not yet validated. However, similarly to Calvo-Henriquez et al. [33], we did not aim to validate a new method of olfactory evaluation in general, but rather to create a fast test for predicting COVID-19, hence RT-PCR was used as a gold standard in assessing diagnostic accuracy. Future studies using validated psychophysical olfactory tests are needed to validate SDOIT as a method of olfactory function assessment. It is also worth noting that although our study performed the psychophysical test with the assistance of an examiner, the simplicity of SDOIT and the labeling of samples with numbers (so the patient does not know what odors are presented) would permit it to be conducted remotely. Subjects’ answers could then be easily obtained using an online tool, such as Google Forms. This approach would increase the availability of the test as a screening method.

## 5. Conclusions

In conclusion, we present a simple, fast, low-cost, and effective SARS-CoV-2 screening strategy based on combining a survey for new-onset OD with a simple disposable odor identification test (SDOIT), which may be useful in identifying individuals suspected of COVID-19 and eligible for isolation and laboratory-testing when possible. Moreover, given the imperfect sensitivity of RT-PCR, a positive result in the proposed screening method could be an indication for retesting in cases where the initial SARS-CoV-2-RT-PCR result was negative. We suggest that when there is enough time and good availability of RT-PCR assay, one may consider the combination of self-reported OD and 10-SDOIT; however, with limited time and resources, the combination of self-reported OD and 8-SDOIT appears to be adequate as a predictor of SARS-CoV-2 positivity. This approach could be especially useful in countries with a high number of COVID-19 cases and limited resources to perform RT-PCR for SARS-CoV-2.

## Figures and Tables

**Figure 1 ijerph-18-10185-f001:**
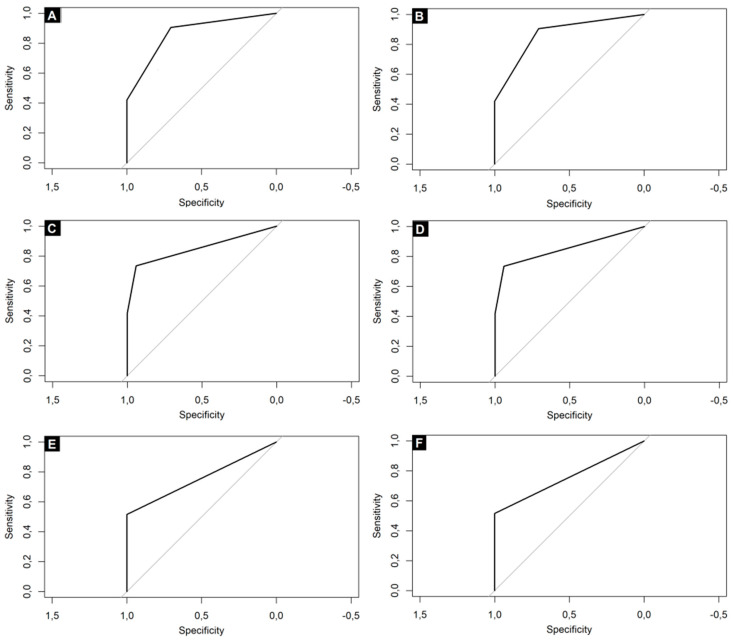
ROC curves for the combination of self-reported OD and: (**A**) 10-SDOIT with optimal cut-off value at 95% (OD defined as 0-9/10 correct answers); (**B**) 10-SDOIT with OD defined as 0-8/10 correct answers; (**C**) 9-SDOIT with OD defined as 0-8/9 identified odors; (**D**) 9-SDOIT with OD defined as 0-7/9 identified odors; (**E**) 8-SDOIT with OD defined as 0-7/8 identified odors; (**F**) 8-SDOIT with OD defined as 0-6/8 identified odors.

**Figure 2 ijerph-18-10185-f002:**
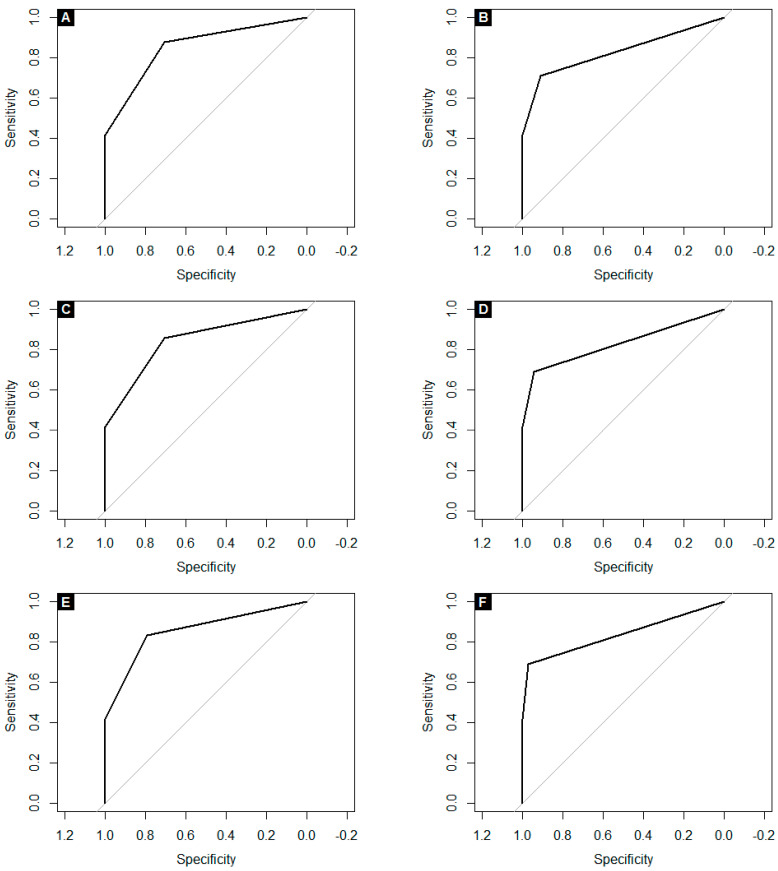
ROC curves for the subset of 75% youngest COVID-19 patients and controls for the combination of self-reported OD and: (**A**) 10-SDOIT with optimal cut-off value at 95% (OD defined as 0-9/10 correct answers); (**B**) 10-SDOIT with OD defined as 0-8/10 correct answers; (**C**) 9-SDOIT with OD defined as 0-8/9 identified odors; (**D**) 9-SDOIT with OD defined as 0-7/9 identified odors; (**E**) 8-SDOIT with OD defined as 0-7/8 identified odors; (**F**) 8-SDOIT with OD defined as 0-6/8 identified odors.

**Table 1 ijerph-18-10185-t001:** Odors and distractors used in the SDOIT.

Title 1	Title 2
cinnamon	honey, vanilla, chocolate
mint	onion, gasoline, garlic
lemon	peach, apple, plum
coffee	tobacco, wine, smoke
clove	grass, garlic, chocolate
rose	green tea, strawberry, cherry
anise	peach, rose, mint
camphor	gas, caramel, onion
alcohol (disinfectant)	gasoline, cucumber, burned rubber
odorless sample	rose, garlic, lemon, mint

**Table 2 ijerph-18-10185-t002:** Comparison of demographic data and smoking status of cases and controls.

Characteristic		Total (*N* = 98)	COVID-19 Patients (*N* = 64)	Control Patients (*N* = 34)
Age, years	mean ± SD	48.4 ± 18.8	52.3 ± 20.9	40.9 ± 10.7
	median (IQR)	47 (32–64)	55 (33–68.5)	40.5 (32–49.8)
	range	20–91	20–91	27–61
Gender, *N* (%)	female	55 (56.1%)	29 (45.3%)	26 (76.5%)
	male	43 (43.9%)	35 (54.7%)	8 (23.5%)
Smoking history	nonsmoker, *N* (%)	64 (65.3%)	35 (54.7%)	29 (85.3%)
	former smoker, *N* (%)	23 (23.5%)	21 (32.8%)	2 (5.9%)
	current smoker, *N* (%)	11 (11.2%)	8 (12.5%)	3 (8.8%)

**Table 3 ijerph-18-10185-t003:** Correlations between self-reported OD and clinical characteristics of COVID-19 patients.

Variable		Self-Reported OD					
		Presence of Self-Reported OD at the Time of the Survey			Presence of Self-Reported OD at Any Time since the Onset of COVID-19		
		Yes (*N* = 21)	No (*N* = 43)	*p*-Value	Yes (*N* = 27)	No (*N* = 37)	*p*-Value
Nasal congestion, *N* (%)	yes	11 (52.4)	10 (23.3)	0.041 ^1^	14 (51.9)	7 (18.9)	0.012 ^1^
	no	10 (47.6)	33 (76.7)		13 (48.1)	30 (81.1)	
Rhinorrhea, *N* (%)	yes	9 (42.9)	10 (23.3)	0.187 ^1^	14 (51.9)	5 (13.5)	0.002 ^1^
	no	12 (57.1)	33 (76.7)		13 (48.1)	32 (86.5)	
Current smoking, *N* (%)	yes	3 (14.3)	5 (11.6)	1 ^2^	3 (11.1)	5 (13.5)	1 ^2^
	no	18 (85.7)	38 (88.4)		24 (88.9)	32 (86.5)	
Former or current smoking, *N* (%)	yes	8 (38.1)	21 (48.8)	0.587 ^1^	8 (29.6)	21 (56.8)	0.058 ^1^
	no	13 (61.9)	22 (51.2)		19 (70.4)	16 (43.2)	
Death, *N* (%)	yes	2 (9.5)	6 (14)	1 ^2^	2 (7.4)	6 (16.2)	0.450 ^2^
	no	19 (90.5)	37 (86)		25 (92.6)	31 (83.8)	
Need for oxygen therapy, *N* (%)	yes	7 (33.3)	21 (48.8)	0.365 ^1^	7 (25.9)	21 (56.8)	0.028 ^1^
	no	14 (66.7)	22 (51.2)		20 (74.1)	16 (43.2)	
Need for ICU stay, *N* (%)	yes	1 (4.8)	5 (11.6)	0.654 ^2^	1 (3.7)	5 (13.5)	0.388 ^2^
	no	20 (95.2)	38 (88.4)		26 (96.3)	32 (86.5)	
Time interval between first positive PCR result and time of the survey, days	Mean ± SD	5.1 ± 4.2	7.6 ± 6.7	0.290 ^3^	6.7 ± 4.8	7.1 ± 7	0.615 ^3^
	Median (IQR)	3 (2–8)	6 (2–12)		5 (2.5–12)	4 (2–12)	
Duration of hospitalisation (excluding deceased patients)	N	19	37	0.862 ^3^	25	31	0.060 ^3^
	Mean ± SD	19 ± 10.6	20.7 ± 14.2		16.6 ± 10.5	23 ± 14.3	
	Median (IQR)	17 (12–23.5)	18 (10–24)		13 (10–19)	18 (15–28)	
MEWS score at the time of the survey	Mean ± SD	0.9 ± 1.3	0.9 ± 1.7	0.837 ^3^	0.7 ± 1.1	1 ± 1.8	0.254 ^3^
	Median (IQR)	0 (0–1)	0 (0–1)		0 (0–1)	1 (0–1)	
Avarage MEWS score	Mean ± SD	0.9 ± 1.5	1 ± 1.3	0.481 ^3^	0.7 ± 1.4	1.1 ± 1.4	0.081 ^3^
	Median (IQR)	0 (0–1)	1 (0–1)		0 (0–1)	1 (0–2)	

^1^ Chi-squared test. ^2^ Fisher test, ^3^ Mann–Whitney test.

**Table 4 ijerph-18-10185-t004:** Correlations between objective OD (according to SDOIT) and quantitative clinical characteristics of COVID-19 patients; Mann–Whitney test.

Variable		SDOIT, % of Correct Answers							
		10-SDOIT		9-SDOIT		8-SDOIT		4-SDOIT	
		Mean ± SD	Median (IQR)	Mean ± SD	Median (IQR)	Mean ± SD	Median (IQR)	Mean ± SD	Median (IQR)
Nasal congestion	yes	72.9 ± 30.4	90 (50–100)	72 ± 33.5	88.9 (55.6–100)	71.4± 33.1	87.5 (50–100)	70.2 ± 40	100 (25–100)
	no	63 ± 32.3	70 (45–90)	62.5 ± 34.6	77.8 (44.4–88.9)	62.8 ± 34.9	75 (37.5–87.5)	55.8 ± 38.1	75 (25–87.5)
	*p*-value	0.188		0.197		0.240		0.081	
Rhinorrhea	yes	73.7 ± 30.4	90 (50–100)	73.1 ± 32.4	88.9 (56–100)	73.3 ± 32.1	87.5 (56.3–100)	69.7 ± 37.8	100 (37.5–100)
	no	63.1 ± 32.2	80 (40–90)	62.5 ± 34.9	77.8 (44.4–88.9)	62.5 ± 35.1	75 (37.5–87.5)	56.7 ± 39.3	75 (25–100)
	*p*-value	0.110		0.152		0.179		0.153	
Current smoking	yes	70 ± 31.2	85 (55–90)	68.1 ± 35.9	83.3 (50–91.7)	68.8 ± 34.7	81.3 (56.4–90.6)	65.6 ± 37.7	75 (43.8–100)
	no	65.7 ± 32.1	80 (47.5–90)	65.3 ± 34.4	77.8 (44.4–88.9)	65.2 ± 34.5	75 (37.5–90.6)	59.8 ± 39.5	75 (25–100)
	*p*-value	0.806		0.837		0.829		0.745	
Former or current smoking	yes	63.1 ± 30.7	80 (40–90)	62.8 ± 32.8	77.8 (44.4–88.9)	62.9 ± 32.8	75 (37.5–87.5)	55.2 ± 40.3	75 (25–100)
	no	68.9 ± 32.9	80 (50–100)	67.9 ± 35.7	77.8 (55.6–100)	67.9 ± 35.8	75 (56.3–100)	65 ± 38	75 (37.5–100)
	*p*-value	0.241		0.259		0.308		0.335	
Death	yes	51.2 ± 22.3	60 (55–60)	50 ± 21.4	55.6 (52.8–58.3)	51.6 ± 23.6	62.5 (46.9–62.5)	40.6 ± 29.7	37.5 (25–50)
	no	68.4 ± 32.5	8 (47.5–90)	67.9 ± 35.3	88.9 (44.4–100)	67.6 ± 35.3	87.5 (37.5–100)	63.4 ± 39.6	75 (25–100)
	*p*-value	0.066		0.048		0.047		0.108	
Need for oxygen therapy	yes	58.9 ± 30.6	6 (47.5–80)	58.3 ± 32.3	61.1 (52.8–80.6)	58.9 ± 32.4	62.5 (46.9–78.1)	49.1 ± 35.7	50 (18.8–75)
	no	71.9 ± 32	90 (47.5–100)	71.3 ± 35.2	88.9 (44.4–100)	70.8 ± 35.2	87.5 (37.5–100)	69.4 ± 39.7	100 (25–100)
	*p*-value	0.032		0.026		0.035		0.013	
Need for ICU stay	yes	68.3 ± 16	60 (60–67.5)	66.7 ± 17.2	61.1 (55.6–66.7)	68.8 ± 17.2	62.5 (62.5–71.9)	58.3 ± 34.2	50 (31.3–87.5)
	no	66 ± 33.1	80 (40–90)	65.5 ± 35.7	77.8 (44.4–88.9)	65.3 ± 35.7	75 (37.5–96.9)	60.8 ± 39.8	75 (25–100)
	*p*-value	0.852		0.700		0.682		0.877	

**Table 5 ijerph-18-10185-t005:** Correlations between objective OD (according to SDOIT) and qualitative clinical characteristics of COVID-19 patients; Spearman correlation.

Variable		SDOIT			
		10-SDOIT	9-SDOIT	8-SDOIT	4-SDOIT
Time interval between first positive PCR result and time of the survey, days	ρ	0.19	0.18	0.19	0.1
	*p*-value	0.123	0.163	0.141	0.447
Duration of hospitalisation (excluding deceased)	ρ	−0.43	−0.42	−0.41	−0.36
	*p*-value	<0.001	0.002	0.002	0.007
MEWS score at the time of the survey	ρ	−0.29	−0.32	−0.32	−0.25
	*p*-value	0.02	0.011	0.011	0.043
Avarage MEWS score	ρ	−0.29	−0.3	−0.29	−0.24
	*p*-value	0.02	0.016	0.018	0.054

**Table 6 ijerph-18-10185-t006:** Comparison of self-reported and objective olfactory function of cases and controls.

Characteristic		COVID-19 Patients (*N* = 64)	Control Patients (*N* = 34)	*p*-Value
Reported smell at the time of maximum deterioration, *N* (%)	normosmia	37 (57.8)	34 (100)	<0.001 ^1^
	hyposmia	21 (32.8)	0 (0)	
	anosmia	6 (9.4)	0 (0)	
Reported smell at the time of the survey, *N* (%)	normosmia	43 (67.2)	34 (100)	<0.001 ^1^
	hyposmia	18 (28.1)	0 (0)	
	anosmia	3 (4.7)	0 (0)	
VAS score of smell deterioration (at the time of maximum deterioration)	mean ± SD	3.4 ± 3.6	0 ± 0	<0.001 ^2^
	median (IQR)	2 (0–7)	0 (0–0)	
VAS score of smell deterioration (at the time of the survey)	mean ± SD	2.6 ± 3.2	0 ± 0	<0.001 ^2^
	median (IQR)	2 (0–5)	0 (0–0)	
SDOIT—detected odors, *N*	mean ± SD	7.5 ± 2.7 (83.5% ± 29.4%)	9 ± 0 (100% ± 0%)	<0.001 ^2^
	median (IQR)	9 (7–9) (100% (77.8%–100%))	9 (9–9) (100% (100%–100%))	
10-SDOIT, correct answers, *N* (%)	mean ± SD	6.6 ± 3.2 (66.3% ± 31.8%)	9.6 ± 0.8 (95.6% ± 8.2%)	<0.001 ^2^
	median (IQR)	8 (4.8–9) (80% (47.5%–90%))	10 (9–10) (100% (90%–100%))	
9-SDOIT, correct answers, *N* (%)	mean ± SD	5.9 ± 3.1 (65.6% ± 34.3%)	8.6 ± 0.7 (95.8% ± 7.7%)	<0.001 ^2^
	median (IQR)	7 (4–8) (77.8% (44.4%–88.9%))	9 (8–9) (100% (88.9%–100%))	
8-SDOIT, correct answers, *N* (%)	mean ± SD	5.3 ± 2.7 (65.6 % ± 34.3%)	7.8 ± 0.5 (97.1% ± 6.2%)	<0.001 ^2^
	median (IQR)	6 (3–7.3) (75% (37.5%–90.6%))	8 (8–8) (100% (100%–100%))	
4-SDOIT, correct answers, *N* (%)	mean ± SD	2.4 ± 1.6 (60.6% ± 39%)	3.9 ± 0.2 (98.5% ± 6%)	<0.001 ^2^
	median (IQR)	3 (1–4) (75% (25%–100%))	4 (4–4) (100% (100%–100%))	

^1^ Fisher test, ^2^ Mann–Whitney test.

**Table 7 ijerph-18-10185-t007:** Comparison of self-reported and objective olfactory function of the subset of 75% youngest COVID-19 patients and controls.

Characteristic		COVID-19 Patients (*N* = 48)	Control Patients (*N* = 34)	*p*-Value
Reported smell at the time of maximum deterioration, *N* (%)	normosmia	28 (58.3)	34 (100)	<0.001 ^1^
	hyposmia	15 (31.2)	0 (0)	
	anosmia	5 (10.4)	0 (0)	
Reported smell at the time of the survey, *N* (%)	normosmia	34 (70.8)	34 (100)	<0.001 ^1^
	hyposmia	12 (25)	0 (0)	
	anosmia	2 (4.2)	0 (0)	
VAS score of smell deterioration (at the time of maximum deterioration)	mean ± SD	3.4 ± 3.7	0 ± 0	<0.001 ^2^
	median [IQR]	2 (0–7)	0 (0–0)	
VAS score of smell deterioration (at the time of the survey)	mean ± SD	2.3 ± 3.1	0 ± 0	<0.001 ^2^
	median [IQR]	0 (0–3)	0 (0–0)	
SDOIT-detected odors, *N*	mean ± SD	7.7 ± 2.5 (85.4% ± 28.1%)	9 ± 0 (100% ± 0%)	<0.001 ^2^
	median (IQR)	9 (8–9) (100% (88.9%–100%))	9 (9–9) (100% (100%–100%))	
10-SDOIT, correct answers, *N* (%)	mean ± SD	7.4 ± 3 (74.2% ± 29.5%)	9.56 ± 0.82 (95.6% ± 8.2%)	<0.001 ^2^
	median (IQR)	9 (6–10) (90% (60%–100%))	10 (9–10) (100% (90%–100%)	
9-SDOIT, correct answers, *N* (%)	mean ± SD	6.6 ± 2.93 (72.9% ± 32.6%)	8.6 ± 0.7 (95.8% ± 7.7%)	<0.001 ^2^
	median (IQR)	8 (5.8–9) (88.9% (63.9%–100%))	9 (8–9) (100% (88.9%–100%))	
8-SDOIT, correct answers, *N* (%)	mean ± SD	5.8 ± 2.6 (72.7 % ± 32.6%)	7.8 ± 0.5 (97.1% ± 6.2%)	<0.001 ^2^
	median (IQR)	7 (5–8) (87.5% (62.5%–100%))	8 (8–8) (100% (100%–100%))	
4-SDOIT, correct answers, *N* (%)	mean ± SD	2.8 ± 1.5 (69.8% ± 36.5%)	3.9 ± 0.2 (98.5% ± 6%)	<0.001 ^2^
	median (IQR)	3 (2–4) (75% (50%–100%))	4 (4–4) (100% (100%–100%))	

^1^ Fisher test, ^2^ Mann–Whitney test.

**Table 8 ijerph-18-10185-t008:** Correlation between self-reported olfactory function and objective test results.

SDOIT Score		Self-Reported Olfactory Function at the Time of the Survey (Normosmia/OD), *N* (%)			VAS Score (Maximum), *N* (%)		
		Normosmia (*N* = 77)	OD (*N* = 21)	*p*-Value	<5 (*N* = 75)	≥5 (*N* = 23)	*p*-Value
10-SDOIT	0–8	25 (32.5)	16 (76.2)	<0.001	27 (36)	27 (60.9)	0.061
	9–10	52 (67.5)	5 (23.8)		48 (64)	9 (39.1)	
9-SDOIT	0–7	22 (28.6)	15 (71.4)	<0.001	24 (32)	13 (56.5)	0.049
	8–9	55 (71.4)	6 (28.6)		51 (68)	10 (43.5)	
8-SDOIT	0–6	6 (7.8)	11 (52.4)	<0.001	7 (9.3)	10 (43.5)	<0.001
	7–8	71 (92.2)	10 (47.6)		68 (90.7)	13 (56.5)	
4-SDOIT	0–3	26 (33.8)	17 (81)	<0.001	26 (34.7)	17 (73.9)	0.002
	4	51 (66.2)	4 (19)		49 (65.3)	6 (26.1)	

**Table 9 ijerph-18-10185-t009:** Results of the ROC analysis.

Classifier	Sensitivity	Specifity	PPV	NPV	AUC
Self-reported OD at the time maximum deterioration	0.42 (CI95% 0.3–0.55)	1 (CI95% 1–1)	1 (CI95% 1–1)	0.48 (CI95% 0.43–0.54	0.71 (CI95% 0.65–0.77)
Maximum VAS	0.64 (CI95% 0.53–0.75)	1 (CI95% 1–1)	1 (CI95% 1–1)	0.6 (CI95% 0.53–0.68)	0.82 (CI95% 0.76–0.88)
10-SDOIT (OD ≥ 1 incorrect)	0.8 (CI95% 0.56–0.78)	0.71 (CI95% 0.62–0.76)	0.84 (CI95% 0.79–0.86)	0.65 (CI95% 0.52–0.77)	0.82 (CI95% 0.74–0.9)
10-SDOIT (OD ≥ 1 incorrect 0-9/10) + self-reported OD	0.91 (CI95% 0.83-0.97)	0.71 (CI95% 0.56–0.85)	0.85 (CI95% 0.79–0.92)	0.8 (CI95% 0.67–0.93)	0.87 (CI95% 0.8–0.93)
10-SDOIT (OD 0-8/10) + self-reported OD	0.77 (CI95% 0.66–0.86)	0.91 (CI95% 0.82–1)	0.94 (CI95% 0.88–1)	0.67 (CI95% 0.58–0.78)	0.86 (CI95% 0.80–0.92)
9-SDOIT (OD ≥ 1 incorrect)	0.77 (CI95% 0.55–0.86)	0.71 (CI95% 0.59–0.97)	0.83 (CI95% 0.77–0.97)	0.62 (CI95% 0.5–0.74)	0.80 (CI95% 0.73–0.88)
9-SDOIT (OD ≥ 1 incorrect 0-8/9) + self-reported OD	0.88 (CI95% 0.83–0.97)	0.71 (CI95% 0.56–0.85)	0.85 (CI955 0.79–0.92)	0.75 (CI95% 0.68–0.93)	0.85 (CI95% 0.79–0.92)
9-SDOIT (OD 0-7/9) + self-reported OD	0.73 (CI95% 0.83–0.97)	0.94 (CI95% 0.56–0.85)	0.96 (CI95% 0.79–0.92)	0.65 (CI95% 0.67–0.93)	0.85 (CI95% 0.79–0.91)
8-SDOIT (OD ≥ 1 incorrect)	0.75 (CI95% 0.61–0.86)	0.79 (CI95% 0.68–0.94)	0.87 (CI95% 0.81–0.96)	0.63 (CI95% 0.53–0.74)	0.82 (CI95% 0.75–0.89)
8-SDOIT (OD ≥ 1 incorrect 0-7/8) + self-reported OD	0.86 (CI95% 0.77–0.94)	0.79 (CI95% 0.65–0.91)	0.89 (CI95% 0.82–0.95)	0.75 (CI95% 0.64–0.88)	0.87 (CI95% 0.81–0.93)
8-SDOIT (OD 0-6/8) + self-reported OD	0.73 (CI95% 0.62–0.84)	0.97 (CI95% 0.91–1)	0.98 (CI95% 0.93–1)	0.66 (CI95% 0.58–0.76)	0.86 (CI95% 0.8–0.92)
4-SDOIT (OD ≥ 1 incorrect)	0.64 (CI95% 0.53–0.75)	0.94 (CI95% 0.85–1)	0.95 (CI95% 0.89–1)	0.58 (CI95% 0.51–0.67)	0.80 (CI95% 0.74–0.87)
4-SDOIT (OD ≥ 1 incorrect) + self-reported OD	0.78 (CI95% 0.67–0.88)	0.94 (C I95% 0.85–1)	0.96 (CI95% 0.91–1)	0.7 (CI95% 0.6–0.8)	0.87 (CI95% 0.82–0.93)

**Table 10 ijerph-18-10185-t010:** Results of the ROC analysis for the subset of 75% youngest COVID-19 patients and controls.

Classifier	Sensitivity	Specifity	PPV	NPV	AUC
Self-reported OD at the time of the maximum deterioration	0.42 (CI95% 0.29–0.56)	1 (CI95% 1–1)	1 (CI95% 1–1)	0.55 (CI95% 0.5–0.62)	0.71 (CI95% 0.64–0.78)
Maximum VAS	0.35 (CI95% 0.21–0.5)	1 (CI95% 1–1)	1 (CI95% 1–1)	0.52 (CI95% 0.47–0.59)	0.68 (CI95% 0.61–0.75)
10-SDOIT (OD ≥ 1 incorrect)	0.73 (CI95% 0.56–0.83)	0.71 (CI95% 0.56–0.91)	0.78 (CI95% 0.7–0.9)	0.65 (CI95% 0.54–0.76)	0.76 (CI95% 0.67–0.86)
10-SDOIT (OD ≥ 1 incorrect 0-9/10) + self-reported OD	0.88 (CI95% 0.77–0.96)	0.71 (CI95% 0.56–0.85)	0.81 (CI95% 0.73–0.9)	0.8 (CI95% 0.67–0.93)	0.85 (CI95% 0.78–0.92)
10-SDOIT (OD 0-8/10) + self-reported OD	0.71 (CI95% 0.58–0.83)	0.91 (CI95% 0.79–1)	0.92 (CI95% 0.83–1)	0.69 (CI95% 0.6– 0.79)	0.83 (CI95% 0.75–0.9)
9-SDOIT (OD ≥ 1 incorrect)	0.71 (CI95% 0.54–0.83)	0.71 (CI95% 0.56–0.91)	0.77 (CI95% 0.69–0.9)	0.63 (CI95% 0.53–0.75)	0.76 (CI95% 0.66–0.85)
9-SDOIT (OD ≥ 1 incorrect 0-8/9) + self-reported OD	0.85 (CI95% 0.75–0.96)	0.71 (CI95% 0.56–0.85)	0.8 (CI955 0.72–0.9)	0.77 (CI95% 0.66–0.91)	0.84 (CI95% 0.77–0.92)
9-SDOIT (OD 0-7/9) + self-reported OD	0.69 (CI95% 0.54–0.81)	0.94 (CI95% 0.85–1)	0.94 (CI95% 0.86–1)	0.68 (CI95% 0.59–0.78)	0.83 (CI95% 0.75–0.9)
8-SDOIT (OD ≥ 1 incorrect)	0.69 (CI95% 0.54–0.81)	0.79 (CI95% 0.65–0.91)	0.82 (CI95% 0.73–0.92)	0.64 (CI95% 0.55–0.76)	0.78 (CI95% 0.69–0.87)
8-SDOIT (OD ≥ 1 incorrect 0-7/8) + self-reported OD	0.83 (CI95% 0.73–0.94)	0.79 (CI95% 0.65–0.91)	0.85 (CI95% 0.76–0.93)	0.77 (CI95% 0.66–0.9)	0.86 (CI95% 0.78–0.93)
8-SDOIT (OD 0-6/8) + self-reported OD	0.69 (CI95% 0.54–0.81)	0.97 (CI95% 0.91–1)	0.97 (CI95% 0.9–1)	0.69 (CI95% 0.6–0.79)	0.84 (CI95% 0.77–0.9)
4-SDOIT (OD ≥ 1 incorrect)	0.54 (CI95% 0.4–0.69)	0.94 (CI95% 0.85–1)	0.93 (CI95% 0.83–1)	0.59 (CI95% 0.52–0.69)	0.75 (CI95% 0.67–0.83)
4-SDOIT (OD ≥ 1 incorrect) + self-reported OD	0.73 (CI95% 0.6–0.85)	0.94 (C I95% 0.85–1)	0.95 (CI95% 0.87–1)	0.71 (CI95% 0.62–0.82)	0.85 (CI95% 0.78–0.92)

## Data Availability

The original, anonymous dataset is available upon request from the corresponding author.

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
