# Peer review of "Simple Disposable Odor Identification Tests for Predicting SARS-CoV-2 Positivity"

_ijerph, 2021, doi:10.3390/ijerph181910185_

Round 1
Reviewer 1 Report
This manuscript is an original work on a Simple Disposable Odor Identification Tests for Predicting SARS-CoV-2 Positivity.
It is an interesting and well-written paper, reinforcing the need for a simple and disposable tool to diagnose and monitor hyposmia during COVID pandemic.
Thus, I strongly recommend its publication, after some minor suggestions:
During Mat & Met: I encourage the authors to include a Figure or a Video showing how their Simple Disposable Odor Identification Test was used (lines 105-107): Ten disposable test paper strips numbered 1 to 10, 9 of which contained well-known pure odorants (commercially 106 available cinnamon, mint, lemon, coffee, clove, rose, anise and camphor oil).
During Discussion, several paragraphs are devoted to explain smell impairment mechanisms in SARS-CoV-2 infection. This is not the objective of this work. I suggest to delete these lines: 310-349.
Author Response
As you suggested
-
During Mat & Met: We have included a Figure showing how the SDOIT was used in Supplementary matherials (Figure S1)
-
We have deleted lines: 310-349 from Discussion
Reviewer 2 Report
This paper presents a series of simple odor tests and relates them to COVID-19 infection. I have a couple of concerns about the differences between the patients and controls - the control group are younger than the patients, which is perhaps not surprising, however, we also know that sense of smell can deteriorate with age. The authors could control for age in their analysis, or alternatively, analyse the data for the youngest 75% of patients and see if there is any difference in conclusions (this subset is more similar to the controls looking at the IQR presented in Table 2). This could also be done for sex and smoking history, which are also different between the two groups. Likewise, the length of time between positive test and the questionnaire has not been accounted for in the analysis, and yet is very variable. Please consider adjusting for it!
Other comments:
- In the methods, please expand and name the "usual descriptive statistics"
- Several large tables are included, but the results are not mentioned in the text. Are they really all required to be included? Add to supplementary, and summarise the interesting ones in a couple of sentences? Currently, they seem pointless.
- Table 6 - what is the bottom line, with no characteristic listed?
- Number of decimal places should be the same in the text and tables, check that all values are 6.62 not 6,62.
- Are the results for the individual odor tests given anywhere? Why not?
- On line 200, should higher be lower?
- Table 7 - middle columns (normosmia/hiposmia/etc), do not add anything and could easily be removed.
- Section 3.4, shouldn't this be earlier in the results section i.e. before 3.3? Then discuss the tests...
- For the tests, use ≥1 incorrect in the tables, rather than 0-9 correct (since this is the wording used in text).
- State upfront what the measure(s) of performance are of interest for the ROC analysis, and present those only. Would be good to have the test and then + self-reported in the next row for easy comparison.
- How is self-reported information combined with the test(s) results? Details should be in methods.
- Figure 2 - combine all ROC curves on one graph for easy comparison.
Minor comments:
- Authors switch between COVID-19 and SARS-CoV-2, this needs to be unified and referred to only in one manner.
- Clinical outcomes - olfactory function is for "sense of smell" not simply smell. Give ratings scale, and then state that this was evaluated at/for different times. Currently reads as though controls were not asked this question?
Author Response
As you suggested:
-
We have analysed the youngest 75% of patients. There was no statistical difference between this subset of patients and controls regarding age. We have compared this subset of patients regarding subjective OD, VAS score and SDOIT results, as well as performed ROC analysis for this subset and controls. The results are shown in the revised manuscript.
-
Regarding sex and smoking history no correlations have been found between these variables and both subjective OD and SDOIT results (as seen in revised tables), therefore we have not performed further analysis with these corrections.
-
There was no significant correletion between self-reported OD and the time between the positive PCR result and the questionnaire, it is reported in the Results in the revised manuscript.
-
We have expanded the term "usual descriptive statistics" in the revised manuscript
-
We have added some of the data included from original manuscript in Supplementary materials, leaving only the most interesting/important in the main manuscript.
-
Table 6 - the bottom line is „4-SDOIT, correct answers, N (%)”
-
We have corrected the number of decimal places to be the same in the text and tables, and checked that all values are 6.62 not 6,62.
-
The results for the individual odor tests have been included in Supplementary materials.
-
On line 201 - it should indeed be „lower” instead of „higher” and it has been corrected.
-
Table 7 – we have removed the middle columns (normosmia/hiposmia/etc)
-
Section 3.3 is before the section 3.4, because it defines different SDOIT models, analysed also in section 3.4.
-
We have replaced 0-9 correct with ≥1 incorrect in the tables
-
We have stated upfront what measures of performance are of interest for the ROC analysis, and presented those only in the main tables, the rest are presented in Supplementary matherials. We have also changed the order in the table, so there would be the test and then the test + self-reported
-
The self reported information is combined with the test(s) results by creating categories as described in Methods (added in the revised manuscript, section 2.5).
-
Figure 2 – combining ROC curves decreased its readability – for that reason we left the figure as in the original manuscript.
For instance, the figure with the combined ROC curves for the combined ROC curves for the age-matched data are attached in PDF file
-
The term COVID-19 is used for the symptomatic disease caused by the SARS-CoV-2 infection (e.g. „inpatients with COVID-19”). The therm „SARS-COV-2 infection” is used for the infection in general, symptomatic or asymptomatic, mainly for screening/epidemiological purposes (e.g. „sole symptom of SARS-CoV-2 infection”, „screening for SARS-CoV-2 infection”, „SARS-CoV-2 positivity” etc.). Otherwise it has been revised.
-
Clinical outcomes:
-
the term „smell” have been replaced by „sense of smell”
-
In cases, as none of the subjects reported an incomplete recovery, the olfactory function at the time of the survey (“current OD”) for the patients reporting complete recovery was classified as “normal” and “0” on the descriptive scale and VAS, respectively, and for the remaining patients classified as equal to the “recent OD”. None of the cases reported any recent changes in the sanse of smell, so the „current OD” and the „recent OD” are equal.
-
